# Anti-TNFα Drugs and Interleukin Inhibitors: Epidemiological and Pharmacovigilance Investigation in COVID-19 Positive Patients

**DOI:** 10.3390/jpm12111770

**Published:** 2022-10-27

**Authors:** Zaira Maraia, Tony Mazzoni, Marco Bruno Luigi Rocchi, Denise Feliciani, Maria Chiara Romani, Giovanna Acciarri, Stefania Rafaiani, Isidoro Mazzoni

**Affiliations:** 1San Benedetto del Tronto Hospital Pharmacy, ASUR Marche AV5, 63074 San Benedetto del Tronto, Italy; 2Department of Pharmacology, University of Camerino, 62032 Camerino, Italy; 3Department of Biomolecular Sciences, University of Urbino, 61020 Urbino, Italy; 4Ascoli Piceno Hospital Pharmacy, ASUR Marche AV5, 63100 Ascoli Piceno, Italy

**Keywords:** anti-TNFα, IL-inhibitors, rheumatoid arthritis, psoriasis, inflammatory bowel disease, COVID-19, SARS-CoV-2, positivity, hospitalization, death

## Abstract

Cytokine patterns and immune activation in patients with Coronavirus 2019 (COVID-19) seem to resemble the case of rheumatoid arthritis (RA), psoriasis and inflammatory bowel disease (IBD). Biological drugs, such as anti-tumor necrosis factor α (TNFα) and interleukin (IL) inhibitors, appear to be protective against adverse outcomes of Severe Acute Respiratory Syndrome Coronavirus 2 (SARS-CoV-2). However, these treatments are associated with an increased risk of secondary infections. The aim of the study was to examine the association between the use of immunomodulatory drugs and the risk of SARS-CoV-2-associated positivity, hospitalization and death compared to other commonly prescribed treatment regimens among patients with immune-mediated inflammatory diseases. Methods: All patients with RA, Psoriasis and IBD were included in this observational analysis and treated with anti-TNFα, IL-inhibitors, Methotrexate (MTX) and Sulfasalazine drugs during the year 2020–2021. The population consisted of 932 patients and demographic, clinical and pharmacological data were analyzed. Results: Although no significant differences were observed between patients treated with biological and synthetic drugs in terms of hospitalization and death, the multivariate logistic model showed that the type of drug influences the possibility of COVID-19 positivity. Conclusions: The results of this analysis support the use of biological drugs and justify further research investigating the association of these biological therapies with COVID-19 outcomes.

## 1. Introduction

Coronavirus 2019 (COVID-19) is the actor of a clinical manifestation called Severe Acute Respiratory Syndrome Corona Virus 2 (SARS-CoV-2). It is well established that the pathogenesis of the disease has three distinct phases. In the first phase, entry of the virus into the body triggers innate and adaptive immune responses to promote elimination of the pathogen and, in most patients, the body responds positively with mild symptoms. The second phase sees the presence of Acute Lung Injury (ALI) characterized by the onset of hypoxemia and bilateral lung infiltrates. In a high percentage of compromised patients infected with COVID-19, the ALI may progress to Acute Respiratory Distress Syndrome (ARDS) characterized by increased vascular permeability associated with the inflammatory state, as well as edema and epithelial cell death. In the third stage of the disease, there is an excessive release of pro-inflammatory cytokines leading to a systemic inflammatory response and an aberrant immune response that contributes to cardiopulmonary collapse and leads to multi-organ dysfunction and death in the most severe patients [1,2]. Cytokine patterns and immune activation in patients with COVID-19 appear to resemble the case of RA, psoriasis and IBD, all of which are autoimmune-based inflammatory diseases. Despite the development of vaccines, monoclonal antibodies and antivirals specifically developed to combat the new coronavirus, for various reasons, there remains a need to expand the therapeutic armamentarium, and the repositioning of already commercially available drugs would make it possible to intervene quickly to reduce COVID-19 mortality.

### 1.1. Biochemistry of TNF-α, IL-12, IL-23 and IL-17

The role of these cytokines in chronic inflammatory diseases, such as RA, psoriasis and IBD, has been extensively studied and anti-TNFα and IL-Inhibitors have led to important changes in clinical practice. TNF-α, or also tumor necrosis factor α, is a cytokine that has pleiotropic effects on various cell types but has been identified as the master regulation of inflammation. TNF-α exerts its actions by interacting with two different receptors: TNFR1 and TNFR2, which initiate signal transduction pathways. Activation of the TNFR1 receptor, which is constitutively expressed in most of our tissues, triggers the formation of several signaling complexes, referred to as complex I, IIa, IIb and IIc, which give rise to distinct cellular responses. The formation of complex I culminates in the activation of nuclear factor κB (NF-κB) and mitogen-activated protein kinases (MAPKs). The formation of complex IIa and IIb leads to the activation of caspase 8, which initiates the apoptotic process, while the formation of complex IIc activates the mixed lineage kinase domain-like protein (MLKL), which induces necroptosis and inflammation. The TNFR2 receptor is only expressed in cells of the immune system and this signaling leads to downstream activation of NF-κB, MAPK and protein kinase B (AKT) [3]. Interleukin-17 (IL-17) is a proinflammatory cytokine with a decisive role in the pathogenesis of various diseases. This cytokine acts by interacting with five different types of receptors distributed widely throughout the body. Depending on the tissue, receptor activation initiates signaling pathways that lead to the activation of transcription factors such as NF-κB, MAPK. IL-17 has synergistic actions with other pro-inflammatory cytokines such as TNF and interleukin-22 (IL-22). IL-17 is a key pleiotropic cytokine in the CD4+ T helper 17 (Th17) cell line [4,5]. Interleukin-12 (IL-12) and Interleukin-23 (IL-23) are heterodimeric cytokines, and this gives them several unique and distinctive characteristics. These cytokines are predominantly proinflammatory and are secreted by activated antigen-presenting cells (APCs), macrophages and dendritic cells. IL-12 stimulates natural killer (NK) cells and leads CD4+ T-cell differentiation towards the T helper 1 (Th1) phenotype while IL-23 induces the Th17 lymphocyte pathway. Abnormal regulation of IL-12 and IL-23 has been associated with immune-mediated diseases, such as psoriasis, psoriatic arthritis, Crohn’s disease and ulcerative colitis [6,7].

### 1.2. Crosstalk between COVID-19 and Immune-Mediated Inflammatory Diseases (IMIDs)

Hyperactivation of the immune response resulting in the excessive release of pro-inflammatory mediators in the lung structures is the main pathological feature of SARS-CoV-2. The hyperactivation of cytokines in COVID-19 appears to be similar to that observed in some chronic autoimmune inflammatory diseases. In the third stage of COVID-19 disease, the dysfunctional immune response produces severe outcomes by initiating a state of hyperinflammation commonly referred to as a ‘cytokine storm’. In this context, the virus induces lymphocyte pyroptosis by activating caspase 1. The APCs trigger the differentiation of naive T-lymphocytes into Th17 lymphocytes, which promote neutrophilic inflammation by activating neutrophils through the release of IL-17. This pathway is further enhanced by the activation of Th1 lymphocytes, which release TNF-α, interleukin-1 (IL-1) and interferon (IFN)-γ. In parallel, RA is also characterized by dysfunctional immunity. Pathological changes are mediated by antibodies directed against autoantigens and inflammation mediated by cytokines produced mainly by Th lymphocytes. A vicious circle is created that is self-amplifying, self-feeding and causes not only joint damage but also important systemic implications. Among these cytokines, TNF-α has been described as the one most strongly involved in the pathogenesis of the disease and has also recently been identified as a key cytokine in the SARS-CoV-2 ‘cytokine storm’ [8,9,10,11]. Psoriasis is a chronic inflammatory dermatosis that results from an interaction between immunological, environmental and genetic factors. This disease involves several cells of the immune system resulting in inflammatory feedback and hyperproliferation of the epidermis. In the initial phase of the disease, plasmacytoid dendritic cells, keratinocytes, NK cells and macrophages secrete TNF-α, INF-γ and IL-1 and activate myeloid dendritic cells. The latter, at the lymph node level, secrete IL-12 and IL-23, which respectively induce the differentiation of naive T lymphocytes into Th1 and Th17 lymphocytes. The released cytokines lead to proliferation of downstream keratinocytes, increased expression of angiogenic mediators and infiltration of immune cells into the injured skin [12,13,14]. The IBD is a chronic condition caused by inappropriate immune activation of the mucosa. The two diseases that constitute IBD are ulcerative colitis and Crohn’s disease. Bacterial components are processed by APCs and presented to T lymphocytes, which differentiate into Th1 and Th17 lymphocytes. TNF-α, IL-17 and interleukin-6 (IL-6) can accumulate intestinal fibroblasts, neutrophils and macrophages. Fibroblasts cause fibrosis responsible for stenosis in the intestine while accumulated neutrophils secrete elastase to induce matrix degradation. Finally, accumulated macrophages in turn release cytokines that induce epithelial and endothelial damage [15,16]. Therefore, it is reasonable to conclude that both SARS-CoV-2 and the above-mentioned IMIDs share a similar mechanistic pathway of aberrant immune response.

### 1.3. COVID-19 Treatment: New Challenges

Effective therapy would depend on the use of immunomodulators to reduce systemic inflammation before it results in multi-organ dysfunction. Immunomodulatory agents that have already been studied include therapies targeting the IL-6 pathway. Medications for rheumatic diseases have been the focus of attention. These treatments are associated with an increased risk of secondary infections. This association raises concern about a reduced immune response to SARS-CoV-2 among these patients as these cytokines play an important role in the defense against pathogens. However, as pointed out earlier, COVID-19 is associated with a hyperinflammatory response [15,17]. Therefore, treatments targeting a hyperactive immune response could exert protection against adverse COVID-19 outcomes. Clinical data to date are, unfortunately, conflicting. The rationale for a possible use of anti-TNFα in COVID-19 stems from clinical studies in rheumatoid arthritis. For example, in RA, anti-TNFα reduces the production of additional cytokines such as IL-1 and IL-6; therefore, blocking TNFα in COVID-19 could reduce these potentially pathogenic cytokines [18,19,20]. Evidence supporting the importance of targeting TNF-α in COVID-19 positive patients comes from observational data on humans extrapolated from three large registries. This showed that being on anti-TNFα therapy for underlying rheumatic disease, psoriasis or IBD, was significantly associated with a lower likelihood of COVID-19-related hospitalization [21,22,23]. However, in a phase 2 clinical trial, no significant association emerged between infliximab and reduced inflammation in patients hospitalized with COVID-19 pneumonia [24].

We asked ourselves the question: “Does the administration of biological drugs in patients with chronic inflammatory diseases give protection against COVID-19 compared to conventional synthetic drugs? Do patients with RA, psoriasis and IBD treated with biologic drugs have a higher or lower risk of COVID-19 than patients treated with synthetic drugs?”. Accordingly, we implemented an active surveillance project to identify cases of COVID-19 that occurred in patients using immunomodulatory therapy and to describe their clinical outcomes. Specifically, the aim of the study was to examine the association between the use of immunomodulatory drugs and the risk of COVID-19-associated positivity, hospitalization and death compared to other commonly prescribed treatment regimens among patients with immune-mediated inflammatory diseases.

## 2. Materials and Methods

During the period 2020–2021, all patients with RA, psoriasis and IBD were included in this retrospective observational analysis. The diagnosis of these diseases was made by the clinic on the basis of its assessments. The patients under analysis are chronic patients and were being treated with anti-TNFα drugs, IL-inhibitors, Methotrexate and Sulfasalazine 12 months before the arrival of the pandemic. In this sample, the outcomes of interest were asymptomatic/symptomatic mild COVID-19 positivity, hospitalization for SARS-CoV-2 and eventual death. Specifically, we collected information regarding COVID-19 test results, hospitalization, treatment, pre-existing conditions and patient outcomes. We conducted descriptive analyses of these patient factors and compared differences between survivors and non-survivors. The survey sample consisted of 932 patients residing in the province of Ascoli Piceno, Marche Region, Italy. COVID-19 positivity was detected with the SARS-CoV-2 real-time polymerase chain reaction (RT-PCR) laboratory test. Data were extrapolated from electronic medical records. Clinical data (IMIDs, COVID-19 diagnosis, hospitalization for ARDS and death due to ARDS) were extracted from the ‘Primary Care’ database available at the hospitals and territorial services of Ascoli Piceno. Pharmaceutical data on the use of the type of drug (biological or synthetic) were extracted using the Apoteke Gold management system, which contains records of all prescriptions from the last 10 years. The latter allows the extrapolation of pharmaceutical prescription data, enabling an epidemiological analysis using the Anatomical Therapeutic Chemical (ATC) classification system. In addition, the prescription of the drugs in question is accompanied by a therapeutic plan drawn up by the specialist, which contains all the information on the patient (name, surname, pathology, prescribed therapy) and is kept in the hospital pharmacy, allowing monitoring and bimonthly dispensing of the drug. For all patients, several variables such as gender, age, diagnosis, drug treatment and presence of comorbidities were recorded on a processing sheet. Patients were stratified according to pharmacological class and type of active substance used. COVID-19 positivity, admission for ARDS and eventual death were reported. The collected data were examined using two types of statistical analysis: univariate and multivariate analysis. First, a descriptive analysis was performed to provide a summary representation of the results of the observations. The indices used are central tendency indices, such as the mean, and dispersion indices, such as the standard deviation (SD). Absolute and relative frequencies have been reported for the categorical variables. The chi-square test was used to investigate in an exploratory manner the independence between the observed frequencies on an ‘A’ variable (COVID-19 positivity, admission for ARDS, outcome of admission for ARDS) and a ‘B’ variable (gender, medication, pathology (Ra or Psoriasis or IBD), comorbidity (presence/absence)). Multivariate logistic regression was used to estimate the probability of COVID-19 positivity, hospitalization and death from ARDS as a function of the other observed variables. Only coefficients with *p* values below the 5% significance threshold were considered statistically significant. This model is a non-linear regression model used when the dependent variable is dichotomous and expresses the log of the ODD, which means the ratio of its probability (p) to the probability that it does not occur (1-p). Statistical processing of the results was performed with Statistical Package for Social Science (SPSS) software, version 23.0 IBM SPSS Company, Armonk, NY, USA, 2018.

## 3. Results

As of 1 January 2020, 484 patients were being treated with biologic drugs and 448 with conventional synthetic drugs. Table 1 shows the general and clinical characteristics of patients treated with anti-TNFα (*n* = 313). The mean age of the considered population (SD) was 57 years (17.5). It was found that most of the selected patients (*n* = 135) were on Adalimumab treatment. Of these patients, 52.6% were male and 47.4% female. In addition, 15.6% (*n* = 21) tested positive for COVID-19 with this treatment regimen, but none were hospitalized or died. With reference to the total sample, only two patients (0.6%) were hospitalized for SARS-CoV-2 and the patient treated with Certolizumab died. However, the latter patient had comorbid heart failure and was slightly older than the survivor (78 vs. 52).

Similarly, Table 2 describes the characteristics of patients treated with interleukin inhibitors (*n* = 171). The mean age (SD) of the total sample was 55 years (16) and 53% were men. The most common disease diagnosis was psoriasis (*n* = 108 [63.2%]) and 18.7% of the total patients had another disease in comorbidity. COVID-19 positivity was found in 14 patients (8.1%), while hospitalization for ARDS occurred in 1 patient (0.6%) but without death. The deceased patient was 88 years old and had heart failure and obesity as comorbid conditions. 

Table 3 shows the characteristics of patients treated with conventional synthetic drugs (*n* = 448). The mean age (SD) of the analyzed population was 65 years (15.6). Most patients (*n* = 298 (66.5%)) were female and 26.3% had comorbidities. Regarding disease diagnosis, among the patients treated with the synthetic drugs, 61.6% presented with RA, 17.2% with psoriasis and 21.2% with IBD. Methotrexate monotherapy (*n* = 353) was the most common treatment regimen compared to Sulfasalazine (*n* = 95). Of the total patients, 17.4% (*n* = 78) tested positive for COVID-19 in asymptomatic/symptomatic mild form. Among the patients treated with Methotrexate, 0.8% (*n* = 3) were hospitalized for ARDS and 0.3% (*n* = 1) died. No patients taking Sulfasalazine therapy at baseline were hospitalized or died. 

In addition, the mean time (SD) of hospitalization was equal to 15.5 days (34.6). The most common comorbidities in the total sample under study were hypertension (*n* = 105 [11.3%]) and diabetes (*n* = 63 [6.7%]).

In an exploratory manner, in the univariate analysis we wanted to observe possible associations between the categorical variables under study, such as COVID-19 positivity, ARDS hospitalization, ARDS hospitalization outcome and the type of drug (biological or synthetic), the type of pathology (AR or psoriasis or IBD), sex and the presence or absence of comorbid pathology. With respect to the diagnosis of COVID-19 positivity, a significant association emerged between patients treated with the biological or synthetic drug (χ^2^ = 4.64; df = 1; *p* = 0.03). Specifically, as can be seen in Figure 1, 17.4% of patients treated with synthetic drugs were COVID-19 positive compared to 12.4% of patients treated with biological drugs. As the percentage of patients treated with synthetic drugs was slightly higher than that of patients treated with biological drugs, the statistical analysis showed that the two variables are not independent but are related to each other. No significant associations were found with the variables sex, pathology and comorbidity.

Focusing on patients who were hospitalized for ARDS, the analysis revealed a significant association with the presence of comorbidity (χ^2^ = 16.07; df = 1; *p* = 0.001). Specifically, all patients who were hospitalized had a comorbidity at the same time (Figure 2). Since two cells had a count of less than 5, as a further check, Fisher’s exact test was applied, from which a significant dependency between the two variables in question emerged (F = 16.06; df = 1; *p* = 0.001). No significant connections emerged with the variables gender, pathology and type of drug.

Similarly, we wanted to study the outcome of patients who were hospitalized for ARDS and data processing revealed, even in this case, a single significant association between the patients who died and the presence of the comorbid condition (χ^2^ = 6.41; df = 1; *p* = 0.01). Compared to the total sample, deaths occurred exclusively in patients with at least one comorbidity (Figure 3). To support this, Fisher’s exact test was performed, from which a significant connection between the two variables under analysis emerged (F = 6.40; df = 1; *p* = 0.01).

In addition, to construct a predictive model, multivariate logistic regression (logit model) was used to estimate the probability of COVID-19 positivity, hospitalization for ARDS and death as a function of the other variables under observation. The purpose of the logit model is to study which variables have the greatest influence in trying to interpret reality. When the β coefficients of the model are positive, this means that the probability of risk increases, conversely when they are negative. Although no significant differences were observed between patients treated with biological and synthetic drugs in terms of hospitalization and death, a particularly interesting finding emerged. The type of drug influences the possibility of COVID-19 positivity. Specifically, patients treated with synthetic drugs have a higher probability of COVID-19 positivity (β = 0.427; *p* = 0.030, OR = 1.532) or conversely, patients treated with biological drugs have a significantly lower probability of COVID-19 (Table 4). 

In multivariate analysis, the only significant factor associated with a higher probability of hospitalization included older age (β = 0.105; *p* = 0.012; OR = 1.111).

## 4. Discussion

It has been hypothesized that therapies targeting cytokines involved in this aberrant inflammatory response may play a role in delaying lung damage in patients with SARS-CoV-2 infection [21,23]. The research and development of new specific anti-SARS-CoV-2 drugs is a lengthy process with the need to demonstrate both efficacy and safety. The repositioning of current therapies represents an interesting option for the management of COVID-19. The therapies to be evaluated are biological drugs with proven efficacy in suppressing hyperinflammation. Pro-inflammatory cytokines, such as TNF-α, are the target of a potential therapeutic strategy in the treatment of severe and critical cases of COVID-19 patients. Immunomodulatory drugs that can currently be used in COVID-19 are those aimed at inhibiting IL-6 and IL-1, thus preventing the effects of activation of the pro-inflammatory cascade. Data for IL-6 and IL-1 inhibition in COVID-19 treatment have been largely positive, suggesting a mortality benefit [25,26]. This observational analysis showed that among patients with immune-mediated inflammatory diseases, COVID-19 positivity was found in 63 patients (13%) treated with biological drugs vs. 78 patients (17.4%) treated with synthetic drugs. The type of drug influenced the possibility of COVID-19 positivity in a statistically significant manner, showing a dependency between the two variables under observation. Similarly, from the logistic model, it was possible to observe that the dependent variable ‘positivity to COVID-19’ has as significance in the type of drug used. The lower probability of an unfavorable outcome of COVID-19 among patients receiving biological drugs prior to SARS-CoV-2 infection has several possible explanations. Although the exact mechanism of the hyperinflammation associated with SARS-CoV-2 remains uncertain, high cytokine concentrations have been associated with organ damage. The administration of biological drugs should reduce the cytokine storm caused by the excessive immune response and inflammation that can occur in COVID-19. This is supported by preclinical studies showing how neutralizing therapies, for example, TNF-α protected SARS-CoV-2 infected mice from tissue damage compared to control mice. A similar response was also observed in sepsis, reinforcing the idea that inhibition of the cytokine pathway may prove useful in patients with COVID-19 by limiting tissue damage. Anti-TNFα therapy has been shown to reduce leukocyte migration, which is evident in COVID-19 pneumonia, as a consequence of reduced expression of endothelial adhesion molecules [20]. The potential role of anti-TNF therapy thus warrants consideration. However, in terms of hospitalization or death, the analysis showed that biological drugs are comparable to synthetic drugs. Evidently, cytokines such as IL-17, IL-23 and IL-12 are not among the most relevant in SARS-CoV-2 related inflammation, but not only that, a lower risk of exposure may help to explain this. Social isolation may have influenced the exposure dose of SARS-CoV-2, which may have influenced the viral load and clinical course of COVID-19. Behavioral variation and the resulting impact on the risk and/or severity of COVID-19 require further investigation. In comparison with the logit model, the univariate analysis showed that the only significant association with hospitalization and death was the presence of comorbidity. The indisputable explanation for this difference is that whereas in univariate statistical analysis one variable is considered at a time and is therefore used in an exploratory manner, in the case of logistic regression one works in a multivariate manner. This means that each meaningfulness is net of the others and is therefore more reliable. The presence of comorbidity had a statistically significant influence on the probability of hospitalization and/or death due to ARDS in accordance with the global findings [27]. Patients with comorbidities are at increased risk of infection and critical situations develop in these patients. SARS-CoV-2 infection becomes harmful when it occurs in a person with comorbidities and the management of these patients with appropriate therapies is an important step for their survival.

This study involves a small number of patients, making further studies necessary to better understand the role of these drugs in COVID-19 positive patients. Other limitations of this analysis refer to the absence of a control and the time-varying treatment patterns of patients for COVID-19. Another limitation attributable to our analysis refers to the type of management system used for data extraction. For example, the presence of comorbidities was only derived based on each patient’s exemption code. Such differences have the potential to introduce bias. However, the strength of the present analysis includes a stratification of the sample, which allowed a more detailed description of the medication among patients with immune-mediated inflammatory disease.

## 5. Conclusions

COVID-19 has had a direct impact on the lives of many people and globally there has been an unprecedented human crisis. Currently, infections caused by the new variants are high despite the administration of several doses of vaccine. There is a need for an effective treatment with a proven safety profile. The results of this observational analysis support the use of biological drugs and warrant further research investigating the association of these biological therapies with COVID-19 outcomes. If this approach proves effective, it could be considered as an extension to non-inpatients at high risk of progression. In addition, the availability of subcutaneously administered formulations could facilitate this type of use. In future studies, it is important to take the time factor into consideration. Therapy with biological drugs may be more effective if administered early. Preventing the development of hyperinflammation might be more beneficial than treating it once it has developed and is often present at the time of hospital admission. In addition, COVID-19 causes hepatic and renal damage, limiting the use of various therapies. In this direction, biological drugs have advantageous pharmacological properties because they are not metabolized in the liver and are not eliminated in the kidney but are broken down into their constituent amino acids. For this reason, there is no need to adjust the dosage in the case of hepatic or renal insufficiency. Although results on several drugs suggest a benefit in patients with COVID-19, only dexamethasone, tocilizumab and sarilumab are recommended in the guidelines. Further clinical studies are needed to ascertain the potential of each drug and the timing of their administration. Although observational data can be used to guide the development of further research, further trials are necessary if one wants to move from a hypothesis to a therapeutic benefit.

## Figures and Tables

**Figure 1 jpm-12-01770-f001:**
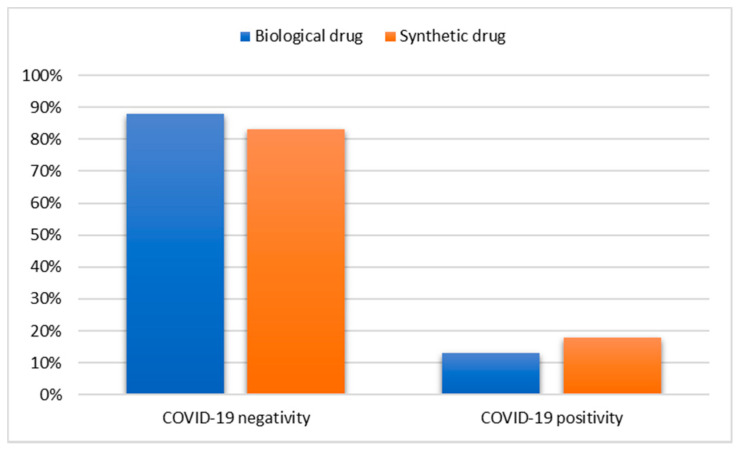
Association between COVID-19 positivity and type of drug used.

**Figure 2 jpm-12-01770-f002:**
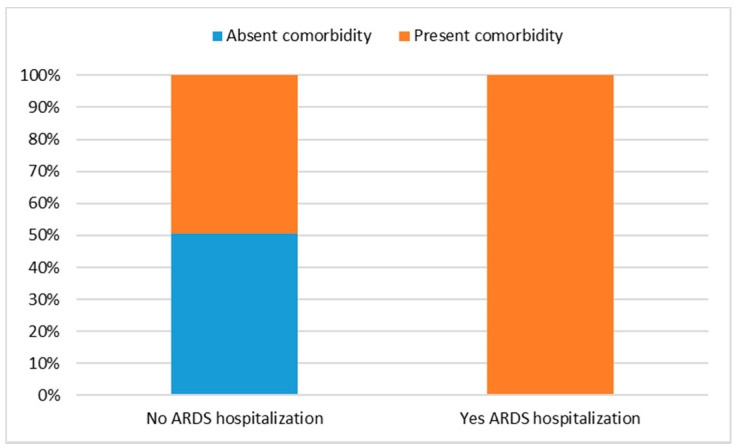
Association between ARDS hospitalization and comorbidity.

**Figure 3 jpm-12-01770-f003:**
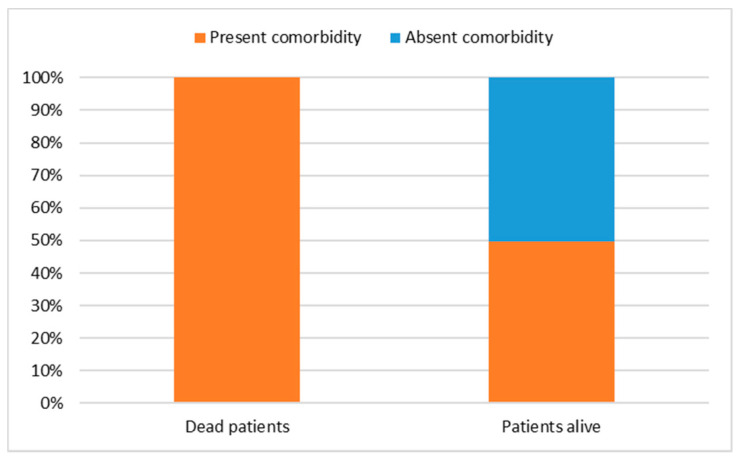
Association between ARDS hospitalization outcome and comorbidity.

**Table 1 jpm-12-01770-t001:** Characteristics of patients treated with anti-TNFα.

	Infliximab	Adalimumab	Certolizumab	Golimumab	Etanercept
Total (%)	28 (100)	135 (100)	39 (100)	39 (100)	72 (100)
Mean age (SD)	53 (15)	55 (19)	58 (17)	56 (13)	60 (18)
Male (%)	16 (57.1)	71 (52.6)	9 (23.1)	15 (38.5)	30 (41.7)
Female (%)	12 (42.9)	64 (47.4)	30 (76.9)	24 (61.5)	42 (58.3)
Rheumatic disease (%)	5 (17.9)	67 (49.6)	37 (94.9)	27 (69.2)	65 (90.3)
Psoriasis (%)	0	21 (15.6)	2 (5.1)	7 (18)	6 (8.3)
IBD (%)	23 (82.1)	47 (34.8)	0	5 (12.8)	1 (1.4)
Comorbid disease (%)	3 (10.7)	29 (21.5)	13 (33.4)	9 (23.1)	20 (27.8)
Positive COVID-19 (%)	8 (28.6)	21 (15.6)	6 (15.4)	6 (15.4)	8 (11.1)
ARDS hospitalization (%)	1 (3.6)	0	1 (2.6)	0	0
ARDS death (%)	0	0	1 (2.6)	0	0

**Table 2 jpm-12-01770-t002:** Characteristics of patients treated with interleukin inhibitors.

	Ustekinumab	Secukinumab	Ixekizumab	Brodalumab	Guselcumab
Total (%)	47 (100)	72 (100)	20 (100)	7 (100)	25 (100)
Mean age (SD)	53 (18)	55 (14)	51 (17)	63 (9)	58 (17)
Male (%)	31 (66)	36 (50)	11 (55)	3 (42.9)	10 (40)
Female (%)	16 (34)	36 (50)	9 (45)	4 (57.1)	15 (60)
Rheumatic disease (%)	15 (32)	38 (52.8)	5 (25)	0	0
Psoriasis (%)	27 (57.4)	34 (47.2)	15 (75)	7 (100)	25 (100)
IBD (%)	5 (10.6)	0	0	0	0
Comorbid disease (%)	7 (14.9)	13 (18.1)	8 (40)	0	4 (16)
Positive COVID-19 (%)	5 (10.6)	2 (2.8)	3 (15)	0	4 (16)
ARDS hospitalization (%)	0	0	0	0	1 (4)
ARDS death (%)	0	0	0	0	0

**Table 3 jpm-12-01770-t003:** Characteristics of patients treated with synthetic drugs.

	Methotrexate	Sulfasalazine
Total (%)	353 (100)	95 (100)
Mean age (SD)	66 (15)	58 (16)
Male (%)	110 (31.2)	40 (42.1)
Female (%)	243 (68.8)	55 (57.9)
Rheumatic disease (%)	276 (78.2)	0
Psoriasis (%)	77 (21.8)	0
IBD (%)	0	95 (100)
Comorbid disease (%)	100 (28.3)	18 (19)
Positive COVID-19 (%)	69 (19.5)	9 (9.5)
ARDS hospitalization (%)	3 (0.8)	0
ARDS death (%)	1 (0.3)	0

**Table 4 jpm-12-01770-t004:** Logistic model with dependent variable ‘COVID-19 positivity’.

	β	S.E.	WALD	df	*p*	Odds Ratio	Lower Confidence Limit	Upper Confidence Limit
Sex	0.071	0.193	0.135	1	0.713	1.073	−0.307	0.449
Age	−0.006	0.006	1.104	1	0.293	0.994	−0.018	0.005
Rheumatic disease			4.284	2	0.117			
Psoriasis	−0.460	0.249	3.415	1	0.065	0.631	−0.948	0.028
IBD	−0.338	0.257	1.731	1	0.188	0.714	−0.840	0.165
Comorbid disease	−0.106	0.227	0.220	1	0.693	0.899	−0.550	0.338
Synthesis drug	0.427	0.197	4.707	1	0.030	1.532	0.041	0.812
Constant	−1.445	0.337	14.689	1	0.000	0.236	−2.185	−0.706

## Data Availability

Data and material are available from the corresponding author.

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
