# Peer review of "Anti-TNFα Drugs and Interleukin Inhibitors: Epidemiological and Pharmacovigilance Investigation in COVID-19 Positive Patients"

_jpm, 2022, doi:10.3390/jpm12111770_

Round 1

Reviewer 1 Report

In the present manuscript, authors have explored " Anti-TNFα Drugs and Interleukin Inhibitors: Epidemiological and Pharmacovigilance Investigation in COVID-19 Positive Patients". The subject is of interest and falls in the topics of Journal of Personalized Medicine. There is poor focus with too many things going on.

After reviewing the manuscript thoroughly, I have the following comments:

Absract: Full form of TNFα, SARS COV-2 is missing. Check these things through the manuscript and correct them.

Figure 1,2 and 3 can be improved for more representation.

“Institutional Review Board Statement: Not applicable. Purely observational studies do not require  registration in Italy.” Provide proof in this regard.

Differentiate the COVID-19, SARS COVID-19 and SARS-COV-2 as you have written these in the manuscript.

There are Covid-19 and COVID-19 in the manuscript. Correct them.

There are a number of diseases that occur due to immune activation. Why did you choose RA, Psoriasis and IBD? 

Reviewer 2 Report

The study attempts to associate background usage of certain csDMARDs/bDMARDs with the risk for COVID-19 infection in patients with RA/IBD/psoriasis. The main drawbacks are: a) a conceptual one, ie the mode of action of these treatments is certainly different when they have been used over several weeks/months (ie, possible immunosuppression and predisposition to COVID) vs. acute administration to lower cytokines levels (ie, protective against severe SARS-CoV2);  b) a methodological one outlined below

1.  Insufficient details are given regarding the detection of cases. Was this based on chart records?  Mild or even moderately severe cases may have been missed.

2. PCR details are given.

3. Too few cases with COVID have been analysed thus lowering the power to detect significant associations. 

4. Statistical section should provide details as how the multivariable models were built and features were selected.

5. Tables should include the average age of the participating under each therapy; this is the single most important confounder for severe COVID-19.

Round 2

Reviewer 2 Report

I have no further comments.